# MWG-UNet: Hybrid Deep Learning Framework for Lung Fields and Heart Segmentation in Chest X-ray Images

**DOI:** 10.3390/bioengineering10091091

**Published:** 2023-09-18

**Authors:** Yu Lyu, Xiaolin Tian

**Affiliations:** School of Computer Science and Engineering, Faculty of Innovation Engineering, Macau University of Science and Technology, Taipa, Macau 999078, China; 2109853gii30002@student.must.edu.mo

**Keywords:** WGAN, organ segmentation, computer auxiliary diagnosis

## Abstract

Deep learning technology has achieved breakthrough research results in the fields of medical computer vision and image processing. Generative adversarial networks (GANs) have demonstrated a capacity for image generation and expression ability. This paper proposes a new method called MWG-UNet (multiple tasking Wasserstein generative adversarial network U-shape network) as a lung field and heart segmentation model, which takes advantages of the attention mechanism to enhance the segmentation accuracy of the generator so as to improve the performance. In particular, the Dice similarity, precision, and F1 score of the proposed method outperform other models, reaching 95.28%, 96.41%, and 95.90%, respectively, and the specificity surpasses the sub-optimal models by 0.28%, 0.90%, 0.24%, and 0.90%. However, the value of the IoU is inferior to the optimal model by 0.69%. The results show the proposed method has considerable ability in lung field segmentation. Our multi-organ segmentation results for the heart achieve Dice similarity and IoU values of 71.16% and 74.56%. The segmentation results on lung fields achieve Dice similarity and IoU values of 85.18% and 81.36%.

## 1. Introduction

With the improvement in medicine, medical image analysis has become an important auxiliary diagnosis and treatment method in medical imaging. The anatomical imaging of organs obtained through medical imaging technologies can objectively reflect tissue structure and pathological change. Medical image segmentation improves efficiency for doctors, from positioning and obtaining evidence to guiding treatment. Deep learning technology has made breakthrough research results in the field of computer vision and image processing. Its powerful feature learning ability has attracted widespread attention. The application of artificial intelligence to medical image processing can not only improve the processing efficiency, but also play an auxiliary role, with doctors subsequently analyzing the condition [1]. As a key branch in medical imaging research, medical image semantic segmentation tasks have always played a pivotal role in clinical diagnosis [2]. The mainstream method studies the segmentation of the lesion area in the frontal image. In this paper, we mainly study the segmentation of multi-position medical images.

Artificial intelligence is increasingly popular with the advancement of massive electronic data and improved technology. In the area of combining artificial intelligence and medical treatment, multiple auxiliary diagnosis systems based on convolutional neural networks have become an inevitable trend in developing new medical models [3], such as the incorporation of positioning, medical image segmentation, and classification. To facilitate accurate and detailed observations of lesions, machine-learning-based automatic recognition and segmentation of medical images can enhance processing efficiency and provide supplementary assistance to doctors to subsequently analyze patients’ conditions [4]. Due to the COVID-19 pandemic, there has been a massive research focus on lung field segmentation with chest X-ray or computed tomography (CT) images for auxiliary diagnosis. The CT images show clearer edges without overlapping organs compared to chest X-ray images. Chest X-ray images are low contrast and have blurred borders for organ segmentation, but chest X-ray images are in common use in hospitals because of their low price and quick image generation. There is a great need to process large amounts of information, which can be achieved through automatic segmentation. The combination of lung field and heart segmentation improves the efficiency, showing the relative positions of the organs and sizes of the lung fields and heart at the same time. The positions of the lung fields and heart also provides information on the potential risk of the heart and lung fields. On the one hand, lung fields and heart segmentation identifies the relative position and sizes of the lung fields and heart to clarify the illness. On the other hand, automatic segmentation improves the speed of doctors’ ability to deal with diseases. The automatic segmentation of chest X-ray images has improved the image review quality and the speed of disease diagnosis. Our interest is in helping patients to detect potential risk with chest X-ray images and improving the speed of doctors’ diagnoses. For these reasons, this paper focus on lung field and heart segmentation with chest X-ray images.

Medical image segmentation is a critical task in the field of medical imaging. It involves the identification and delineation of specific structures or regions of interest within medical images captured using techniques such as magnetic resonance imaging (MRI), CT, or ultrasound scans. The purpose of segmentation is to extract accurate boundaries or contours of anatomical structures, tumors, lesions, or other areas of clinical significance. Accurate segmentation plays a crucial role in various medical applications, including disease diagnosis, treatment planning, surgical guidance, and monitoring of disease progression. It allows clinicians to analyze and quantify specific regions, measure volumes, track changes over time, and assist in making informed decisions for patient care. However, medical image segmentation is a challenging task due to several factors. Firstly, medical images often exhibit low contrast, noise, and variations in intensity, making it difficult to distinguish between different structures or tissues. Additionally, the shapes and sizes of anatomical structures can vary significantly across patients and even within the same patient, further complicating the segmentation process. To address these challenges, various segmentation techniques have been developed, ranging from traditional methods to more advanced deep learning approaches. Traditional techniques include threshold, region-based methods, active contour models, and graph cuts. These methods rely on image characteristics, statistical information, or prior knowledge to delineate structures.

GANs are capable of generating synthetic data that closely resemble real data, making them useful in various applications such as image synthesis, video generation, and text generation. GANs can learn from unlabeled data, which eliminates the need for manually labeled training examples. This makes GANs flexible and adaptable to a wide range of domains and datasets. In novel data creation, GANs have the potential to generate entirely new and unseen data samples. This can be valuable for creative tasks, generating unique artwork or exploring uncharted areas of data distribution. GANs can be used to augment existing datasets to increase their size and diversity. This helps improve the generalization and performance of machine learning models trained on limited data. GANs can be employed for domain adaptation, where they learn to generate data from a source domain to match the statistics of a target domain. This facilitates transferring knowledge learned from one domain to another.

GAN training can be challenging and unstable at times. It involves a delicate balance between the generator and discriminator networks and finding this equilibrium can be difficult. GANs may suffer from issues like mode collapse, where the generator fails to explore the entire data space, or vanishing gradients. GANs are prone to skipping modes or failing to capture the complete data distribution. This means that certain aspects or modes of the data may not be adequately represented by the generated samples. Assessing the quality of GAN-generated samples objectively is still an open research problem. While subjective evaluation by human observers is often used, developing reliable quantitative metrics for GAN evaluation remains a challenge. Training GANs can be computationally demanding and time consuming, requiring powerful hardware resources like GPUs (graphics processing units) and extensive training iterations to achieve desirable results. The realistic nature of GAN-generated data raises ethical concerns about potential misuse, such as deep-fake technology or generating deceptive content. Ensuring the responsible use of GANs is an ongoing area of concern and research.

In recent years, deep learning approaches, particularly convolutional neural networks (CNNs), have shown remarkable success in medical image segmentation. CNN-based architectures, such as U-Net, SegNet, and DeepLab, have demonstrated high accuracy and efficiency in segmenting diverse medical structures with minimal manual intervention. These models can learn hierarchical features directly from the images and capture complex patterns for accurate segmentation. Supervised classifier learning is also employed. Refs. [5,6] introduced the concept of generative adversarial networks, with impressive results on image-generation-quality benchmarks. This approach involves the interplay between a generator and a discriminator, enabling comprehensive training of the discriminator to achieve autonomous segmentation [7]. Furthermore, deep learning techniques incorporating multi-modal fusion are extensively employed in medical imaging for accurate medical image segmentation [8]. In the book *Intelligent Data Analysis for Biomedical Application* [9], the authors [10] utilized machine learning to successfully classify myocardial ischemia using delayed contrast enhancement techniques of magnetic resonance imaging. Additionally, addressing the domain shift problem, a new unsupervised domain adaptive framework called the collaborative image and feature adaptive algorithm (SIFA) [11] was proposed and found to be effective. In another study, Ref. [12] directly extracted features from the frequency data of vibration signals and evaluated the performance of feature learning from the original data, and spectrum and time–frequency combined data. This approach successfully applied deep learning in feature extraction for machine-based diagnosis. Furthermore, Ref. [13] introduced a graph convolution method that utilized multi-resolution pools to capture local and contextual features. Their method enabled the learning of spatially related features in irregular domains within a multi-resolution space. A graph-based convolution method employing product neural networks was proposed for position and direction classification, resulting in improved pose parameter estimation and segmentation performance. Another innovative approach, presented by [14], is the multi-receiving domain CNN (MRFNet) method. MRFNet employs an encoder–decoder module (EDM) with sub-net, providing optimal receiving fields for each sub-net and generating context information at the functional map level. MRFNet exhibits exceptional performance across all three medical image datasets.

Edge accuracy is often a key aspect and challenge in segmentation [15]. One proposal suggests utilizing a parameter model with correlation probability density to describe the integration method in the largest posterior form [16]. In 2019, Hiroki Tsud introduced a method that utilizes generative adversarial networks (GANs) with multiple functions for cell image segmentation [17]. This method demonstrates improved segmentation accuracy compared to the traditional pix2pix approach [18]. The field of medical imaging technology generates vast amounts of data. To address this, a heterogeneous framework for multi-core learning based on support vector machines (SVMs) was proposed by [19]. They investigated the flexibility of this method in comparison to using SVMs and other classifiers to process single features which can enhance the learning ability of the Math Kernel Library (MKL) algorithm. Additionally, Nilanjan Dey proposed several medical applications based on meta-heuristics for segmentation [20]. CT images are commonly employed not only for analyzing X-ray films but also for examining the internal structure of the heart, necessitating segmentation of the heart region itself [21,22]. Due to the complexity and significance of large blood vessels attached around the atria and ventricles, a team led by Lohendran Baskaran devised a method for multi-organ segmentation from coronary computed tomography angiography images [23].

In contrast, Olivier Ecabert presented a model that encompasses four heart chambers and interconnected large blood vessels [24]. Avi Ben-cohen introduced a method utilizing a fully convolutional network (FCN) with global context and local plaque level analysis based on super-pixel sparse classification for the detection of liver metastases [25]. Addressing the complementary nature of salient edge and object information, Ref. [26] proposed an edge navigation network that employs a three-step approach to simultaneously incorporate and model these two complementary pieces of information within a single network. The experimental results demonstrate improved performance, particularly in scenarios involving rough object boundaries. Additionally, Ref. [27] proposed a multi-layer densely connected super-resolution 3D network with training guided by generative adversarial networks (GANs). This approach enables fast training and reasoning that outperforms other popular deep learning methods by achieving four times higher image resolution restoration while running six times faster. Furthermore, a novel system has been developed that utilizes CT scans to generate positron emission tomography (PET) virtual images. This system offers the potential for cardiac sarcoidosis evaluation [28]. Ref. [29] proposed multiple tasks deep learning model for detection of peripherally inserted central catheter (PICC) which aids in accurate identification and placement of catheters for medical procedures. Zhongrong Wang proposed pixel-wise weighting-based fully convolutional neural networks for left ventricle segmentation in short-axis MRI [30]. LF-SegNet [31] is a fully convolutional encoder-decoder network designed for lung fields segmentayion from chest radiographs images which assists in automated analysis and diagnosis of respiratory conditions for enabling more efficient medical interventions.

While significant progress has been made in organ segmentation using deep learning methods, there are still several gaps and challenges that exist. The availability of large-scale annotated datasets for organ segmentation is limited. Creating accurate and comprehensive annotations requires significant time and expertise, leading to a scarcity of labeled data. This constraint hinders the development and evaluation of robust models. The organs can exhibit significant variations in shape, size, and appearance across individuals and even within the same individual due to factors such as pathology or imaging artifacts. Existing algorithms often struggle to handle this variability, leading to sub-optimal segmentation results. Certain organs may have indistinct boundaries or overlap with neighboring structures, making their precise delineation challenging. Algorithms need to effectively handle these ambiguous cases and accurately differentiate organ boundaries from surrounding tissues. Deep learning methods primarily rely on data-driven learning without explicitly incorporating prior anatomical knowledge. Integrating prior knowledge, such as anatomical atlases or spatial constraints, into the segmentation process could improve accuracy and consistency.

The major contributions of this paper are outlined as follows.

A designed U-Net with an SE block called AR-UNet, which takes advantages of the attention mechanism to enhance the segmentation accuracy of the generator so as to improve the performance.By applying the AR-UNet as the generator of the MWG-UNet structure, the proposed model can both promote accurate lung field segmentation and enhance the stability in model training.The proposed MWG-UNet is comprehensively evaluated on the JRST and Shenzhen Hospital datasets and achieves the optimal performance for most evaluation metrics except IoU.

The remainder of this paper is organized as follows. Section 2 introduces the proposed method in detail. Section 3 presents the experimental results. Section 4 presents a discussion of our method and others. In Section 5, conclusions are drawn.

## 2. Materials and Methods

The methodology includes three parts: the flaws of GANs, Wasserstein GAN, and the improvement of WGAN. The flaws of GANs introduces the shortcomings of GANs from the perspective of mathematical principles. Wasserstein GAN introduces the working principles of WGAN and the improvement compared with GAN. The improvement of WGAN introduces the improvements that have been made based on WGAN.

### 2.1. GANs

Generative adversarial networks (GANs) are generative models and their training is in the form of a confrontational game. Equation (Equation 1) is the objective loss function of the confrontation.
(1)$$\begin{array}{c} V\left(D,G\right)=E_{x∼P_{r}}\left[logD\left(x\right)\right]+Ez∼P_{g}\left[log\left(1-D\left(G\left(z\right)\right)\right)\right] \end{array}$$
where $P_{r}$ is the true sample distribution, and $P_{g}$ is the sample distribution of the generator.
(2)$$\begin{array}{c} D\left(x\right)=\frac{P_{r}\left(x\right)}{P_{r}\left(x\right)+P_{g}\left(x\right)} \end{array}$$

Equation (Equation 2) represents the simplified optimal discriminator. It is derived by setting the derivative of Equation (Equation 1) to 0 and expresses the loss function that captures the relative proportion of the true distribution and the probability of generating the distribution.

One challenge with GAN training is the need to avoid over-training the discriminator. If the discriminator becomes too proficient, it hampers the reduction in the generator’s loss function during the experimental phase. Another issue encountered in GANs is the limited diversity of the generated samples. In scenarios where the true sample distribution and the generator’s sample distribution exist as low-dimensional manifolds within a high-dimensional space, the probability of their overlap being negligible approaches 1. Consequently, regardless of how dissimilar they are, the JS divergence remains constant, resulting in the generator’s gradient approaching (approximately) 0 and effectively disappearing.

In summary, GANs encounter challenges related to the discriminator’s proficiency and the lack of diversity in generated outputs. Mitigating these challenges is crucial to achieving better performance and diversity in GAN-based architecture. The Wasserstein distance is defined as follows:(3)$$\begin{array}{c} KL\left(P_{g}∥P_{r}\right)=2JS\left(P_{r}∥P_{g}\right) \end{array}$$

There are two significant issues with this equivalent minimization objective. Firstly, it aims to minimize the KL divergence between the true distribution and the generator’s distribution, while simultaneously maximizing the JS divergence between them. This approach is flawed, since the Kullback–Leibler (KL) divergence is not a symmetrical measure, leading to intuitive absurdity and numerical gradient instability.

Furthermore, the generator tends to prioritize generating repetitive and safe samples over diverse samples, exacerbating the challenges within GAN frameworks. These fundamental problems can be attributed to the unreasonable nature of the equivalent optimization distance metric and the generator’s loss function, which result in unstable gradients and imbalanced penalties between diversity and accuracy.

To address the latter concern, a proposed solution involves introducing noise to both the generated and real samples. Intuitively, this noise “diffuses” the original low-dimensional manifolds into the entire high-dimensional space, compelling them to have a noticeable overlap. Once an overlap exists, enabling the presence of a gradient, the JS divergence can effectively operate. Notably, as the two distributions become closer, the diffusion of overlap reduces the JS divergence. However, despite these improvements the quest for a quantitative indicator to measure training progress remains unresolved.

### 2.2. Wasserstein GAN

The Wasserstein distance, introduced in WGAN, addresses the problem of gradient disappearance in theory. Unlike KL divergence and JS divergence, which exhibit abrupt changes and can only be the largest or smallest values, the Wasserstein distance maintains smoothness. When researchers aim to optimize parameters using the gradient descent method, KL and JS divergences fail to provide gradients altogether, while the Wasserstein distance remains capable of providing a gradient.

Similarly, in a high-dimensional space, if two distributions lack overlap or have a negligible overlap, KL and JS divergences cannot accurately represent the distance or offer gradients. In contrast, the Wasserstein distance can provide meaningful gradients in such scenarios. The key advantage of the Wasserstein distance over KL divergence and JS divergence is its ability to reflect the distance between two distributions even when there is no overlap.

By utilizing the Wasserstein distance instead of JS divergence, stable training and progress indicators can be simultaneously achieved. The Wasserstein distance is defined as follows:(4)$$\begin{array}{c} W\left(P_{r},P_{g}\right)=inf_{r\in \prod (P_{r},P_{g})}E_{(x,y)∼r}[∥x-y∥] \end{array}$$

Formula (4) represents the marginal distribution of each distribution for the true sample distribution and the sample distribution of the generator. For each possible joint distribution γ, the expected value of the distance of the sample under the joint distribution γ can be calculated. The Wasserstein distance cannot be drawn directly and the loss function becomes the following formula:(5)$$\begin{array}{c} W\left(P_{r},P_{g}\right)=\frac{1}{K}sup_{{∥f∥}_{L}} \end{array}$$

When the Lipschitz continuity condition limits the maximum local variation in a continuous function, a parameter can be constructed. When the last layer of the discriminator network is not a non-linear activation layer, the loss function will approximate the distance between the true distribution and the generated distribution. Due to the excellent nature of the Wasserstein distance, there will be no problem with the disappearance of the generator gradient. The loss functions of the generator and discriminator are as follows:(6)$$\begin{array}{c} minL_{D}\left(P_{r+\epsilon },P_{g+\epsilon }\right)=-E_{x∼P_{r+\varepsilon }}\left[logD^{x}\left(x\right)\right]-E_{x∼P_{g+\varepsilon }}\left[log\left(1-D^{x}\left(x\right)\right)\right] \end{array}$$

According to Equation (Equation 6), the smaller the value the better the training. After the improvement of WGAN, there are several improvements compared to the original GAN algorithm:The sigmoid function is not applicable in the discriminator. GAN’s discriminator performs a two-classification task of true and false. The discriminator in WGAN approximates the Wasserstein distance. So there is no need for the sigmoid function.The loss of the generator and discriminator does not take the log function.It limits all parameters of the neural network to no more than a certain range.It completely solves the problem of GAN training instability; there is no longer a need to carefully balance the training level of the generator and the discriminator.During training there is a value like cross-entropy to indicate the progress of the training.

### 2.3. The Overall Framework of MWG-UNet

The basic framework of our proposed method, called MWG-UNet, is shown in Figure 1. The architecture of MWG-UNet contains a discriminator for distinguishing real and fake data and a generator for making the distribution of generated samples close to the real data. In the generator of MWG-UNet, we use an improved U-Net called ARU-Net, with a squeeze and excitation (SE) block. The generator is responsible for creating synthetic data samples and attempting to generate realistic outputs that can deceive the discriminator. The discriminator is tasked with distinguishing between the real and fake data samples produced by the generator. The results of the generator and the real data are input into the discriminator to distinguish the real data from the fake data to finally realize accurate segmentation. The parameter from the discriminator updates the generator to improve the performance of the generator.

Figure 2 shows the details of the designed AR-UNet as the generator of MWG-UNet. AR-UNet is U-Net transformed with an SE block and residual connection. The structure of AR-UNet comprises an encoder and decoder. The encoder captures the high-level features of an input image and reduces its spatial dimensions, while the decoder recovers the spatial information and generates a segmentation mask by upsampling the encoded features to match the original input resolution. For every step of downsampling, we take 3 × 3 convolution layers to extract the features and the SE block for boosting the discriminated power of each channel, improving both the accuracy and efficiency in various computer vision tasks. After that, we use a 2 × 2 max pooling operation to extract high-dimensional features. On the contrary, for upsampling we use a 2 × 2 upsampling operation to enlarge the feature image. Then, we use the same method as every step of downsampling before max pooling for image processing. There are four steps for the upsampling operation and downsampling operation. In order to map the feature, we use a 1 × 1 convolution layer at the last layers. The skip connection enables the direct flow of low-level spatial information from the encoder to the decoder for facilitating precise localization and fine-grained details in the output and helps mitigate the problem of information loss during max pooling, improving the overall segmentation accuracy of U-Net. For the sake of improving the efficacy in medical image segmentation to obtain spatial features, we take non-linear addition at the final step with a 1 × 1 convolution layer for the output.

The discriminator of MWG-UNet is shown in Figure 3. In every step of the discriminator, after every two convolutions, the downsampling operator is implemented. After four steps of the max pooling operation, the final output is given by the fully connected layer. Through a series of convolutional or fully connected layers, the discriminator learns to discern subtle patterns and features. The discriminator guides the generator’s learning process by providing feedback on how well it can deceive the discriminator. In this study, a simple discriminator is used for accurate segmentation and saving unnecessary computation.

Unstable training is a common issue with GANs. Although WGAN has made significant strides in achieving stability, there are instances where it generates poor samples or struggles to converge. The introduction of Wasserstein GAN shifts the measurement of probability distributions in GANs from f-divergence to Wasserstein distance, resulting in improved stability during training and generally higher-quality generated outputs. However, WGAN relies on a weight clipping strategy to enforce the Lipschitz constraint on the critic, which can lead to undesired behavior during the training process. To address this limitation, this paper proposes a different truncation pruning strategy known as gradient penalty. This strategy penalizes the gradient’s norm with respect to the critic’s input. By incorporating gradient penalty, the training of WGAN becomes more stable, and the quality of the the generated images improves.

In the realm of medical image segmentation, U-Net and transformed U-Net are commonly used methods. Our approach aims to enhance the performance and robustness by combining the concepts of U-Net and adversarial networks. The paper introduces the addition of Gaussian noise to the generated images and utilizes batch normalization in the discriminator to achieve higher-quality generated results with improved stability. We use the structure of Wasserstein GAN for the adversarial network. In the generator, we use the UNet structure for feature extraction. UNet, as the generator, has an encoding path for context and extracting features and a decoding path for upsampling to recover the spatial information lost during the encoding phase. The combination of WGAN and UNet improves the performance on lung segmentation with regards to the Jaccard similarity and Dice similarity metrics. Details on the value of the combination are shown in Section 4.

This paper proposes a new architecture called the multi-tasking Wasserstein generative adversarial network U-shape network (MWG-UNet). Multi-tasking refers to the ability of a system or an individual to perform multiple tasks simultaneously or in rapid succession. The goal of multi-task learning is to improve the generalization and performance on each individual task by jointly optimizing the model’s parameters across multiple tasks. In the context of computer systems and artificial intelligence, multi-tasking refers to the capability of a program or a model to handle and execute multiple tasks concurrently. For chest X-ray medical images, multi-tasking improves the efficiency of diagnosis with the relative positions and the shapes of the heart and lung fields. The combination of WGAN and UNet improves the performance of lung segmentation with regards to Jaccard similarity and Dice similarity.

## 3. Results

The Results and Discussion address two areas: image pre-processing and segmentation results. They introduce the results of using different generative adversarial networks. The training of the methods is conducted on a Linux system version 18.04, GPU environment, cuda10.2, cudnn7.6.5, and Python environment 3.8.

### 3.1. Data Pre-Processing

The original image for transformation and organs segmentation in Figure 4. The Japanese Society of Radiological Technology (JSRT) [32] is a public dataset of chest X-ray images with accurate annotation for lung fields and heart masks. JSRT encompasses 154 nodule and 93 non-nodule 12-bit gray-scale images with high resolution 2048 × 2048. As we know, machine learning needs massive amounts of data to optimize the model and avoid overfitting. The medical images used in the experiment are provided by Shenzhen No.3 Hospital in Shenzhen [33]: 340 normal X-ray images and 275 abnormal X-ray images. The total number of original images is 862. We use geometric transformations and rotation for data augmentation to increase the data to avoid gradient explosion. In this paper, the adaptive histogram equalization (CLAHE) operation is used for image enhancement. Data augmentation is a common method used in image processing for medical image segmentation and classification because public datasets with accurate annotation are commonly small, which can cause overfitting. Common data augmentation techniques can be divided into basic image manipulation and deep learning approaches. The heart part and the lung part of all pictures are marked by the doctor. All algorithms use 70% as the training sample and 30% as the test sample and all images are converted to 512 × 512 pixels PNG format.

### 3.2. Evaluation Metrics

As a binary question for organ segmentation, five measures are displayed, including Dice similarity, IoU, recall, precision, and F1 score. Equations (7)–(11) show their mathematical formulae. These metrics are based on the true positive (TP), true negative (TN), false positive (FP), and false negative (FN) classification results. In image segmentation, Dice similarity, also known as the Dice coefficient, has commonly been used as a metric to quantify the similarity or overlap between two sets or binary masks. The IoU, also called the Jaccard similarity, reflects the similarity of the segmentation with deep learning and the ground truth, which is the most intuitive indicator to compare the performance with other methods. The Dice similarity, also known as the Dice coefficient, is a statistical measure used to quantify the similarity or overlap between two sets or binary masks. It calculates the ratio of twice the intersection of the sets to the sum of their sizes, providing a value between 0 and 1, where 1 indicates perfect overlap and 0 indicates no overlap. It is commonly used in image segmentation tasks to evaluate the accuracy of predicted segmentation masks by comparing them with ground truth masks. Higher Dice similarity scores indicate better agreement between the predicted and ground truth masks. Recall, also known as sensitivity or true positive rate, measures the proportion of actual positive instances correctly identified by the model. Precision represents the proportion of predicted positive instances that are actually true positives. The F1 score is the harmonic mean of recall and precision, providing a balanced measure that considers both metrics. The equations of these five metrics are presented below.
(7)$$\begin{array}{cc} Dice & =2*TP/(FP+FN+2*TP)\\ \; & =2*(true*pred\_score)/[true*true+pred\_score*pred\_score] \end{array}$$
(8)$$\begin{array}{c} IoU=(TP)/(TP+FP+FN) \end{array}$$
(9)$$\begin{array}{c} Recall=TP/(TP+FN) \end{array}$$
(10)$$\begin{array}{c} Precision=TP/(TP+FP) \end{array}$$
(11)$$\begin{array}{c} F1score=2*TP/[(TP+FN)+(TP+FP)]=2*(ture*pre)/(true+pred) \end{array}$$

The subdivision indices of the results generated after the image segmentation are shown in Table 1. This table compares the results of five metrics obtained by different networks of improvised lightweight deep CNNs [34], UNet++ and HardNet [35], UNet and EfficientNet [36], AlexNet and ResNet [37], GAN, and WGAN and MWG-UNet. The results of other models are directly cited from the corresponding literature. The values without citation are trained by our machine. According to the results, it is found that the proposed MWG-UNet achieves the best results in terms of Dice, precision, and F1 score compared with other algorithms. In particular, the Dice similarity, precision, and F1 score of the proposed method outperform other models, reaching 95.28%, 96.41%, and 95.90%, respectively, and the specificity surpasses the sub-optimal models by 0.28%, 0.90%, 0.24%, and 0.90%. However, the value of IoU is inferior to the optimal model by 0.69%. In the image segmentation, the value of Dice demonstrates the average performance and the value of IoU tends to measure the worst performance. Our results are not as good as the optimal model for some difficult cases, which indicates that the model’s generalization ability needs to be improved.

However, single-organ segmentation demonstrates less information for diagnosis. This paper also expands on a new area for lung field and heart segmentation to improve the efficiency of diagnosis. The relative positions of the heart and lung fields can demonstrate some heart or lung disease. The segmentation of heart and lung fields intuitively displays the sizes and positions, allowing direct diagnosis of related diseases. Our lung field and heart segmentation results achieve Dice similarity and IoU values of 71.16% and 74.56%. The segmentation result of lung fields achieve Dice similarity and IoU values of 85.18% and 81.36%. Figure 5 shows the original medical images and a comparison of the ground truth and transformed images for the heart and lung fields. Column (a) shows the original chest X-ray images. Column (b) demonstrates the ground truth for the chest X-ray images with the segmentation of the heart and lung fields. A comparison with column (c) and column (d) demonstrates an increase in smoothness and clarity. However, the accuracy for each organ in the lung field and heart segmentation has decreased much more than in single-organ segmentation. Figure 5 shows examples of partial segmentation results. Figure 5a shows different non-processed and pre-processed medical images for organ segmentation for the heart and lungs. Figure 5b shows the ground truth of heart and lung segmentation with multiple colors for different parts of organs. The blue part is the left lung. The green part is the right lung of the lung field. The red part shows the heart segmentation. Figure 5c shows the segmentation with WGAN. Figure 5c is the segmentation result using our method, MWG-UNet. Comparing columns (c) and (d), the result of the segmentation in column (d) is slightly better than column (c).

## 4. Discussion

In the era of big data, artificial intelligence is touching all fields, including medical image processing. Deep learning methods for automatic organ segmentation improve the efficiency of diagnosis. However, organ segmentation presents challenges due to irregular shapes, occlusions, image artifacts, and the limited availability of annotated data. Developing robust and efficient algorithms capable of handling these challenges is crucial for enhancing clinical decision making and improving patient care outcomes. Several deep learning methods have been developed for organ segmentation in medical imaging. U-Net is a popular architecture known for its encoder–decoder structure and skip connections, enabling precise localization. Mask R-CNN combines object detection with instance segmentation, providing accurate organ delineation. FCN utilizes fully convolutional layers to generate pixel-wise predictions. V-Net extends U-Net with 3D convolutions, suitable for volumetric data segmentation. DenseNet employs dense connectivity patterns to enhance feature reuse and gradient flow. Attention-based models like DeepLab utilize spatial attention mechanisms to focus on relevant regions. These methods leverage the power of deep neural networks in capturing intricate patterns and have shown promising results in organ segmentation tasks.

In this paper, we proposed a new method, called MWG-UNet, for medical image segmentation. The proposed method uses the designed AR-UNet as the generator and several convolutional and fully connected layers as the discriminator. AR-UNet combines the U-Net architecture for semantic segmentation with an SE block that captures channel-wise dependencies. The SE block re-calibrates feature maps, enhancing the discriminative power. This integration improves both localization accuracy and feature representation in U-Net, making it more effective for organ segmentation tasks. MWG-UNet with U-Net combines the WGAN framework for stable training of GANs with the AR-UNet architecture for semantic segmentation. This integration enables the generation of realistic and high-quality segmentation masks by leveraging the benefits of both an improved loss function and the ability to capture spatial information and preserve fine details.

The training data used were from JSRT and Shenzhen Hospital, with 862 images. The medical images from Shenzhen No.3 Hospital in Shenzhen provide 340 normal X-ray images and 275 abnormal X-ray images. JSRT encompasses 154 nodule and 93 non-nodule 12-bit gray-scale images with high resolution 2048 × 2048. To handle the limited dataset, we use geometric transformations and rotation as data augmentation to increase the data to avoid gradient explosion. The evaluation metrics are Dice similarity, IoU, recall, precision, and F1 score. As mentioned above, our results for the Dice similarity, precision, and F1 score of the proposed method outperform other models, reaching 95.28%, 96.41%, and 95.90% and the specificity surpasses the sub-optimal models by 0.28%, 0.90%, 0.24%, and 0.90%. The result of the IoU is slightly lower than the optimal method. The IoU demonstrates a worse performance in training. In further research, improving the ability of generalization is an important step to upgrade our algorithm. Our lung field and heart segmentation results have Dice similarity and IoU values of 71.16% and 74.56% for the heart segmentation. The segmentation results of the lung fields have Dice similarity and IoU values of 85.18% and 81.36%.

Based on the above discussion, it is shown that the proposed MWG-UNet is a lung field and heart segmentation model which takes advantage of the attention mechanism to enhance the segmentation accuracy of the generator so as to improve the performance. Although the proposed method presents sub-optimal results on the IoU value, there is still space for further improvement. Firstly, our novel method uses WGAN and AR-UNet for lung segmentation in chest X-ray images and outperforms other methods. Secondly, we proposed an improved U-Net with an SE block which is responsible for creating synthetic data samples and attempting to generate realistic outputs. Lastly, we use multiple task learning for multiple-organ segmentation to improve the efficiency of diagnosis with the relative positions and sizes of the organs.

## 5. Conclusions

Medical image segmentation is a critical task in the field of medical imaging. In this paper, we focus on overcoming the difficulties of traditional generative adversarial networks, such as a vanishing gradient and the diversity of generators. Wasserstein GAN was introduced to address the instability of GAN training and the diversity of generated samples. We conduct an in-depth mathematical analysis to highlight the disparities between the mathematical foundation and the practical training of Wasserstein GAN. The article proposes a method, MWG-UNet, for lung field segmentation with chest X-ray images. This method was applied to a public dataset of lung and heart segmented X-ray images. The results for Dice similarity, precision, and F1 score of the proposed method outperform other models, reaching 95.28%, 96.41%, and 95.90%, and the specificity surpasses the sub-optimal models by 0.28%, 0.90%, 0.24%, and 0.90%. The result of the IoU is slightly lower than the optimal method. Meanwhile, the paper explores the frontiers for lung field and heart segmentation with X-ray images. Although the accuracy is lower than for single-organ segmentation, the relative positions of the heart and lung fields provide various information for disease diagnosis. The lung and heart segmentation results achieve Dice similarity and IoU values of 71.16% and 74.56% for the heart segmentation. The segmentation result of the lung fields achieve Dice similarity and IoU values of 85.18% and 81.36%. In future work, we will focus on increasing the accuracy of the lung field and heart segmentation results and on speeding up the training and testing.

## Figures and Tables

**Figure 1 bioengineering-10-01091-f001:**
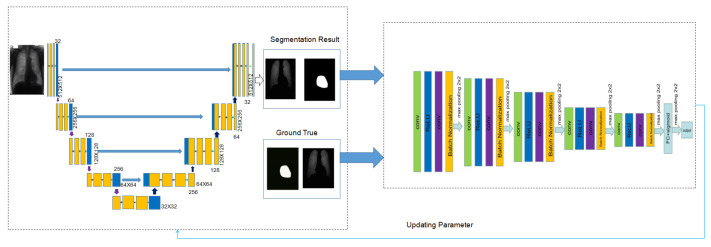
Structure of MWG-UNet.

**Figure 2 bioengineering-10-01091-f002:**
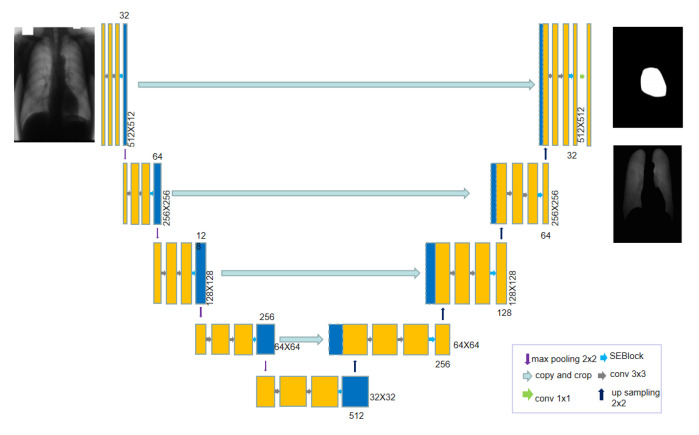
The architecture of AR-UNet.

**Figure 3 bioengineering-10-01091-f003:**
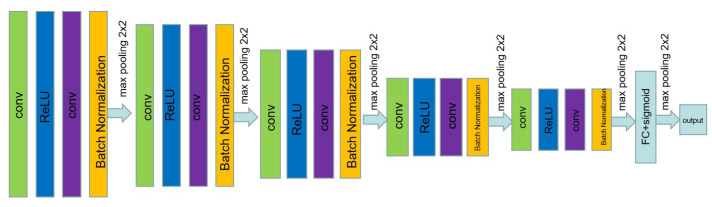
The discriminator of MWG-UNet.

**Figure 4 bioengineering-10-01091-f004:**
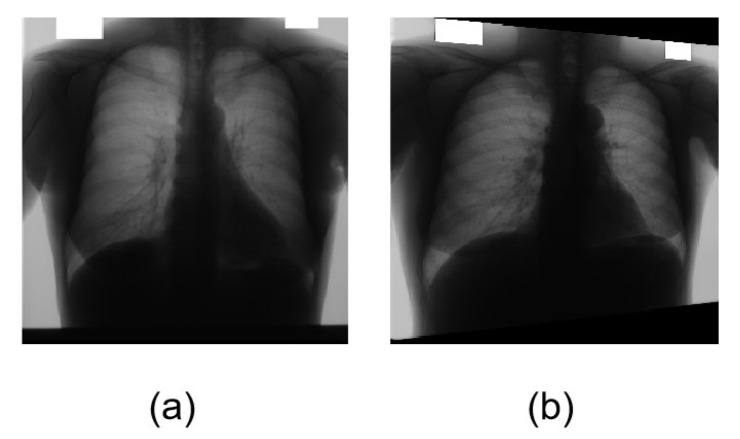
Chest X-ray images. (**a**) The original chest X-ray medical image. (**b**) The figure is transformed with rotation.

**Figure 5 bioengineering-10-01091-f005:**
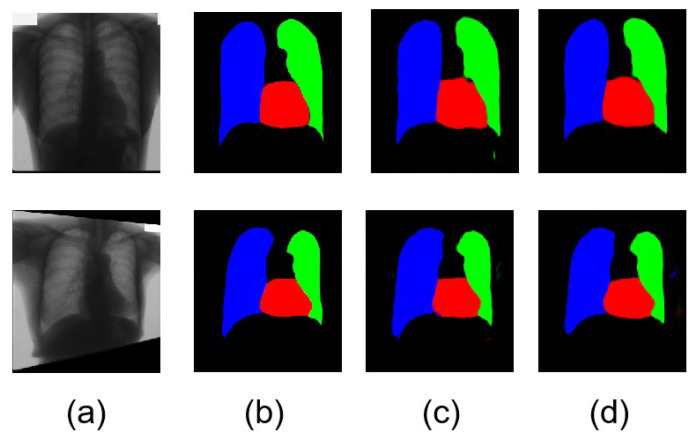
(**a**) The chest X-ray images and segmentation results. (**b**) Ground truth for chest X-ray images. (**c**) Segmentation result using WGAN. (**d**) Segmentation result using our improved method, MWG-UNet.

**Table 1 bioengineering-10-01091-t001:** Segmentation results of different methods for lung fields.

Method	Dice	IoU	Recall	Precision	F1 Score
Improvised lightweight deep CNN	90.64%	86.53%	/	/	/
UNet++ and HardNet	95%	93%	96.5%	94%	95%
UNet and EfficientNet	95%	90%	/	94%	/
AlexNet and ResNet	93.56%	88.07%	/	/	/
GAN	89.16%	86.75%	90.9%	80.69%	85.49%
WGAN	92.23%	89.96%	91.95%	88.93%	90.41%
MWG-UNet	95.28%	92.31%	97.40%	94.24%	95.79 %

## Data Availability

The source code and learned models are available from the authors upon reasonable request.

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
