# Peer review of "MWG-UNet: Hybrid Deep Learning Framework for Lung Fields and Heart Segmentation in Chest X-ray Images"

_bioengineering, 2023, doi:10.3390/bioengineering10091091_

Round 1

Reviewer 1 Report

1. The results and discussion section are presented in an abstract way. It needs proper organization of contents from environment setup till presentation and analysis of results with appropriate metrics and plots. 

2. Novelty of the proposed lung segmentation need to be emphasized properly. 

3. Literature review of the current methods is not sufficient. Authors need to survey the recent methods and include in the discussion. The following are some of the works that can be discussed. 

  • Automated Semantic Segmentation of Chest X-ray images using Deep Learning Model
  • A deep ensemble network for lung segmentation with stochastic weighted averaging

More specific comments. 

1.  Results section needs to include Loss-accuracy plots to know the detail of convergence of the proposed model.

2.  Authors have not reported enough details on the data augmentation part. 

3.  The gaps in the existing works need to be consolidated and discussed at the end of Intro section.

4. Research contributions need to be given at the end of the Intro section.

5. Statistics of the dataset need to be tabulated. 

6. Performance comparison need to be studied with valid segmentation assessment metrics. 

  •  

Moderate changes required

Author Response

Response to Reviewer 1 Comments

Point 1: The results and discussion section are presented in an abstract way. It needs proper organization of contents from environment setup till presentation and analysis of results with appropriate metrics and plots. 

Response 1: Thank you for your suggestion. The second edition would improve it.

Point 2: Novelty of the proposed lung segmentation need to be emphasized properly. 

Response 2: The novelty of lung segmentation is combing WGAN and UNet for Chest X-Ray image segmentation is at the end of introduction.

Point 3: Literature review of the current methods is not sufficient. Authors need to survey the recent methods and include in the discussion. The following are some of the works that can be discussed. 

  • Automated Semantic Segmentation of Chest X-ray images using Deep Learning Model
  • A deep ensemble network for lung segmentation with stochastic weighted averaging

Response 3: Thank you for your advice. These two methods would be mention in the manuscript. 

Point 4:  Results section needs to include Loss-accuracy plots to know the detail of convergence of the proposed model.

Response 4: Thank you for your advice. I am sorry for missing Loss-accuracy plots. After revising it based on the revision comments, I realized on the last day that I had forgotten this plot. At the first day of receiving comment, I write down all comment in my notebook. At the last day, I check it and find out I miss it. Could you please give me another chance for next revision.

Point 5: Authors have not reported enough details on the data augmentation part. 

Response 5: Thank you for your advice. The detail on the data augmentation will be mentioned in the next edition.

Point 6: The gaps in the existing works need to be consolidated and discussed at the end of Intro section.

Response 6: Thank you for your advice.  The gaps is discussed at the end of Intro section.

Point 7: Research contributions need to be given at the end of the Intro section.

Response 7: Thank you for your advice. The contributions is given at the end of Intro section.

Point 8: Statistics of the dataset need to be tabulated. 

Response 8: Thank you for your advice. I only know how many patients get sick. The detail of dataset shows in the Data Pre-Processing.

Point 9: Performance comparison need to be studied with valid segmentation assessment metrics. 

Response 9: Thank you for your advice. We use five metrics for comparing performance, including dice similarity, IoU, recall, precision and F1 score.

Reviewer 2 Report

In order to overcome the drawbacks and limitations of GAN on Multi-organ Segmentation in Chest X-ray, the manuscript proposed an combines the advantages of GAN and UNet to propose a new method called MWG-UNet.

The manuscript proceeds in a straightforward, logical manner, and has been well written. However, the Results and Discussion section have been poor written, which should be heavily improved.

1.     The authors stated that they have performed multi-organ segmentation, so what’s the multi-organ?

2.     Authors should discuss the current literatures about multi-organ segmentation.

3.     How is the detail about the algorithm of image segmentation? And how the MWG-UNet has been used in the algorithm? The Methods section has been written very poor.

4.     How many images has been used and processed in the study? Is the number enough for the study?

5.     How is the efficient of the algorithm? Authors should carefully discuss the proposed algorithm, and should compare it thoroughly with other algorithms.

Author Response

Response to Reviewer 2 Comments

Point 1: The authors stated that they have performed multi-organ segmentation, so what’s the multi-organ? 

Response 1: Thank you for your question. In the manuscript, multi-organ means that lung fields and heart segmentation synchronously.

Point 2:  Authors should discuss the current literatures about multi-organ segmentation.

Response 2: Thank you for your advice. After revising it based on the revision comments, I realized on the last day that I had forgotten it. At the first day of receiving comment, I write down all comment in my notebook. At the last day, I check it and find out I miss it. Could you please give me another chance for next revision.

Point 3:  How is the detail about the algorithm of image segmentation? And how the MWG-UNet has been used in the algorithm? The Methods section has been written very poor. 

Response 3: I am sorry for my poor methods section. I will write more detail of our algorithm in the second edition.

Point 4:  How many images has been used and processed in the study? Is the number enough for the study?

Response 4: Thank you for your comment. We have 862 images. After data augmentation, we have 3448 images. It is enough for our study.

Point 5:  How is the efficient of the algorithm? Authors should carefully discuss the proposed algorithm, and should compare it thoroughly with other algorithms. 

Response 5: Thank you for your suggestion. The second edition will compare the efficiency with other algorithm.

Thank you for your advice to improve our manuscript. I am sorry that the detail of our algorithm don’t show in the manuscript. In the second edition, manuscript will have detail about algorithm and efficiency comparison.

Hope you have a good day! Thank you very much.

Reviewer 3 Report

  1. In the title, the word “Segmentation” is written twice. It’s a bit unclear why the authors need to distinguish between lung and multi-organ segmentation.
  2.  The abstract should focus more on what novel contributions were proposed in this manuscript rather than highlighting what it studies.
  3. Typos: “changes.The”, “process.To”, “models.The”, “Where P_ris”, “P_g is”
  4. “[…] provides huge help […]” please use a formal communication.
  5. Please improve the presentation of the motivation for performing this study.
  6. “MRI, CT”, please note that all abbreviations must be introduced before the first usage.
  7. “The flaws of GAN”, the subtitles must be set more professional. If the GANs had so many flows, then nobody would use them.
  8. Please check that all notations are introduced in this manuscript.
  9. Not all authors read the manuscript. There are quite many typos and formulations that show the experience of a young researcher. No experienced researcher was involved in preparing the manuscript.
  10. Section 2.3 title: “proposal of MWG-UNet”. Please pay more attention to details.
  11. “from f divergence to Wasserstein distance”?
  12. Section 2.3, What exactly is the proposed architecture? There aren’t that many details presented. What exactly is the novel contribution of this manuscript? The presentation is quite general like we should read some other article(s) to get all the details.
  13. “DGX2 Station Xeon CPU E5-2698 memory machine equipped Tesla V100”. The authors need to describe the system used to train the model, not mention the type of machine. One is interested in the system description so one can reproduce these results.
  14. Table 1, please check the state-of-the-art, there should be some other methods that must be introduced in the table. Moreover, please provide some details on how these results were obtained.
  15. The authors need to provide more details regarding the dataset and the network training.

General comments:

  1. The manuscript is missing a lot of details regarding the proposed method, which is barely described in a single page.
  2. The presentation must be improved.
  3. The experimental evaluation is not convincing.

The presentation must be improved. There are quite many typos and informal formulations. 

Author Response

Response to Reviewer 3 Comments

Point 1: In the title, the word “Segmentation” is written twice. It’s a bit unclear why the authors need to distinguish between lung and multi-organ segmentation. 

Response 1: Sorry for my mistake. I want to show the contribution of my both work for lung segmentation and multi-organ segmentation. The new title is “MWG-UNet: Hybrid Deep Learning Framework for Multi-organ Segmentation in Chest X-ray Images”.

Point 2: The abstract should focus more on what novel contributions were proposed in this manuscript rather than highlighting what it studies.

Response 2: Thank you for your advice. I will focus on my novel contribution in the abstract.

Point 3: Typos: “changes.The”, “process.To”, “models.The”, “Where P_ris”, “P_g is” 

Response 3: I am so sorry for these mistakes. All of these will be corrected.

Point 4:  “[…] provides huge help […]” please use a formal communication.

Response 4: Thank you for your advice.This sentence has been changed to “The medical images segmentation improves efficiency for doctors from positioning and obtaining evidence to guiding treatment”.

Point 5: Please improve the presentation of the motivation for performing this study.

Response 5: Thank you for your advice. I will improve my presentation of the motivation.

Point 6: “MRI, CT”, please note that all abbreviations must be introduced before the first usage.

Response 6: Thank you for your advice. All abbreviations are introduced before the first usage.

Point 7: “The flaws of GAN”, the subtitles must be set more professional. If the GANs had so many flows, then nobody would use them.

Response 7: Thank you for your advice. I am sorry for so unprofessional expression. My idea wants to introduce that original GAN structure is not suitable for our dataset. The Wasserstein GAN will be better try for our transformation. I change the subtitle to “GAN”.  

Point 8: Please check that all notations are introduced in this manuscript. 

Response 8: Thank you for your advice. I will check all notation in this manuscript.

Point 9: Not all authors read the manuscript. There are quite many typos and formulations that show the experience of a young researcher. No experienced researcher was involved in preparing the manuscript.

Response 9: I am so sorry. My supervisor read my manuscript and point out some mistakes. My supervisor works differ from my research. My supervisor is 74 years old and ready for retire. The authors of this manuscript are my supervisor and me. Actually, this is my first manuscript for my study. I am very sorry again for my unprofessional manuscript. I will try my best to improve my manuscript.

Point 10: Section 2.3 title: “proposal of MWG-UNet”. Please pay more attention to details.

Response 10: Sorry for my mistakes. The new subtitle is “The details of MWG-UNet”.

Point 11: “from f divergence to Wasserstein distance”?

Response 11: The f divergence is equivalent to a divergence "factory" that must be specified for the generation function f (x) in the formula. The f divergence will generate the specified measurement algorithm based on the specific content corresponding to the generating function f(x). The Wasserstein distance is used for Wasserstein GAN. The formula used f divergence in Wasserstein GAN, we called it Wasserstein distance.

Point 12: Section 2.3, What exactly is the proposed architecture? There aren’t that many details presented. What exactly is the novel contribution of this manuscript? The presentation is quite general like we should read some other article(s) to get all the details.

Response 12: I am sorry that detail of my proposed architecture is not introduced. In the second edition, I will add more detail.

Point 13: “DGX2 Station Xeon CPU E5-2698 memory machine equipped Tesla V100”. The authors need to describe the system used to train the model, not mention the type of machine. One is interested in the system description so one can reproduce these results.

Response 13: Thank you for your advice. The type of machine will be deleted. The system used to train the model will be mentioned. The sentence has been changed to “The training of the methods is conducted on Linux system version 18.04,GPU environment, cuda10.2, cudnn7.6.5 and Python environment 3.8”.

Point 14: Table 1, please check the state-of-the-art, there should be some other methods that must be introduced in the table. Moreover, please provide some details on how these results were obtained.

Response 14: Thank you for your advice. I will add some methods in the table. The result in the table some of them are written in their paper. Some of them are recurrent their methods in the dataset.

Point 15: The authors need to provide more details regarding the dataset and the network training.

Response 15: Thank you for your advice. The second edition will introduce more details regarding the dataset and the network training.

General comments:

  1. The manuscript is missing a lot of details regarding the proposed method, which is barely described in a single page.
  2. The presentation must be improved.
  3. The experimental evaluation is not convincing.

Thank you for your all advice and comment for this manuscript. According this comment, we realize the shortcoming of our manuscript. We are so appreciate for your work with careful reading. We will add more detail that we miss before. The presentation need to be improved in the future. I am so sorry that shows my unprofessional expression. The experimental evaluation will increase for comparison. At last, I am very appreciate for your comment and so sorry for my poor manuscript.

Hope you have a good day! Thank you!

Round 2

Reviewer 1 Report

Authors have incorporated my suggestions 

Minor

Author Response

Dear reviewer 1,

I hope this email finds you well. I am writing to express my sincere gratitude for taking the time to review my paper titled “MWG-UNet: Hybrid Deep Learning Framework for Lung Segmentation and Multi-organ Segmentation in Chest X-ray Images”. Your valuable feedback and insightful comments have been instrumental in enhancing the quality and clarity of my work.

Thank you once again for your invaluable contribution to my work. I look forward to your future insights and the opportunity to collaborate again.

With heartfelt appreciation,

Yu Lyu

Reviewer 2 Report

the revised manuscript is almost good enough for publish. I have only one question:  

If the "Multi-organ" the manuscript described, refers to only two organs: lung and heart, then actually two human organs are not typically referred to as "multi-organ", usually refers to conditions or diseases involving more than two organs.  From this point of view, authors should revise both the title and the contents of the manuscript.

Author Response

Dear reviewer 2,

I hope this email finds you well. I am writing to express my sincere gratitude for taking the time to review my paper titled “MWG-UNet: Hybrid Deep Learning Framework for Lung Segmentation and Multi-organ Segmentation in Chest X-ray Images” .Your valuable feedback and insightful comments have been instrumental in enhancing the quality and clarity of my work.

The title and content with multi-organ all change to "lung fields and heart".

Thank you once again for your invaluable contribution to my work. I look forward to your future insights and the opportunity to collaborate again.

With heartfelt appreciation,

Yu Lyu 

Reviewer 3 Report

The manuscript still contains several typos, but the quality was a bit improved. I would recommend the authors to provide a more thorough presentation of their answers to the provided comments as they need to clearly present what was modified in the manuscript.

The manuscript still contains several typos.

Author Response

Response to Reviewer 3 Comments

Thank you for your comment. I realized there are some points with a more thorough presentation of their answers. Some points with detail answers as follows:

Point 1: The abstract should focus more on what novel contributions were proposed in this manuscript rather than highlighting what it studies.

Response 2: Thank you for your advice. I will focus on my novel contribution in the abstract.

The novel contribution as follows:

1.A designed U-Net with SE block called AR-UNET, which takes advantages of the attention mechanism to enhance the segmentation accuracy of generator so as to improve the most performance.

2.By applying the AR-UNET as the generator of MWG-UNet structure, the proposed model can both promote accurate lung fields segmentation and enhance the stability in model training.

3.The proposed MWG-UNet is comprehensively evaluated on JRST and ShenZhen hospital dataset which performs optimal performance for most evaluation metrics except IoU.

Point 2: Please improve the presentation of the motivation for performing this study.

Response 5: Thank you for your advice. I will improve my presentation of the motivation. Our idea for this study is to automatic segmentation for massive data of chest X-Ray images, because chest X-Ray is a common test for lung and bone disease with initial examination and physical examination for low price. I am sorry for my poor presentation of motivation in the paper. In the second edition, we improve the motivation in the introduction. The motivation in the paper as follows:

Artificial intelligence is increasingly popular with the advancement of massive electronic data and improved technology. In the area of combination of artificial intelligence and medical treatment, multiple auxiliary diagnosis systems based on convolution neural networks has become an inevitable trend in developing new medical models, such as corporation of positioning, medical images segmentation and classification. To facilitate accurate and detailed observation of lesions, machine learning-based automatic recognition and segmentation of medical images can enhance processing efficiency and provide supplementary assistance to subsequent doctors for analyzing patients’ conditions. Due to epidemic COVID-19, massive researchers focus on lung fields segmentation with Chest X-Ray or Computed Tomography(CT) images for auxiliary diagnosis. The CT images show more clarify edges without overlapped organ than Chest X-Ray images. Chest X-Ray images images are with low contrast and blurred border for organ segmentation, but Chest X-Ray images are in common use in hospital because of low price and quick release images. There is a greater need to process large amounts of information through automatic segmentation. Lung fields and heart segmentation improve the efficiency to show the position of relative organ and size of lung fields and heart at the same time. The position of lung fields and heart also promotes potential risk of heart and lung fields. On the one hand, lung fields and heart segmentation performs the relative position and size of lung fields and heart to clarify the illness. On the other hand, automatic segmentation improves the speed of doctors to deal with diseases. The automatic segmentation of Chest X-Ray images has improved images review quality and speed for disease diagnosis. Our interest is helping patients for detect potential risk with Chest X-Ray images and improving speed for doctors to diagnosis. According to these reasons, this paper focus on lung fields and heart segmentation with Chest X-Ray images.

Point 3: Section 2.3, What exactly is the proposed architecture? There aren’t that many details presented. What exactly is the novel contribution of this manuscript? The presentation is quite general like we should read some other article(s) to get all the details.

Response 12: I am sorry that detail of my proposed architecture is not introduced. In the second edition, I will add more detail.

The architecture is shown in section 2.3. The detail of proposed methods shows in the three figure and content of section 2.3. The novel of this manuscript presents in the point 1. Again, I am so sorry for my uncompleted detail of proposed methods. In the section 2.3, we demonstrate the overlook of proposed method of main architecture and detail of discriminator and generator.

Point 4: Table 1, please check the state-of-the-art, there should be some other methods that must be introduced in the table. Moreover, please provide some details on how these results were obtained.

Response 14: Thank you for your advice. The results of other models are directly cited from corresponding literature. The values without citation are trained by our machine. There are three methods add in the second edition. There are two of them which recommended by other reviewers. They published from 2022 to 2023.

Point 5: The authors need to provide more details regarding the dataset and the network training.

Response 5: Thank you for your advice. The second edition will introduce more details regarding the dataset and the network training.

The detail of network training is in the section 2.3. The dataset used in the proposed methods as follows.

 Japanese Society of Radiological Technology (JSRT) is public dataset of Chest X-Ray imagess with accurate annotation for lung fields and heart masks. JSRT encompasses 154 nodule and 93 non-nodule 12-bits gray scale images with high resolution 2048x2048. As we known, machine learning needs massive data to optimize the model and avoid over fitting.  The medical images used in the experiment are provided by Shenzhen No.3 Hospital in Shenzhen for 340 normal X-Ray images and 275 abnormal X-ray images. The total number of original images is 862.

Point 6:The manuscript still contains several typos.

Response 6: Sorry for my mistakes. I reread our paper for revision and check the words with red line, and revise some typo including “hospital”,”Respectively”...

Dear reviewer3,

Thank you for reviewing our paper. I hope this email finds you well. I am writing to express my sincere gratitude for taking the time to review my paper titled “MWG-UNet: Hybrid Deep Learning Framework for lung fields and heart Segmentation in Chest X-ray Images” Your valuable feedback and insightful comments have been instrumental in enhancing the quality and clarity of my work.

I am genuinely thankful for your expertise and the effort you put into reviewing my work. Your meticulous attention to detail and thoughtful observations have undoubtedly enriched the final version of our paper, contributing to its overall strength and impact. Your input has been invaluable to me as a researcher and will undoubtedly contribute to the advancement of my field.

Once again, thank you for your time and commitment to providing such comprehensive feedback. I genuinely appreciate your support and guidance throughout the review process. Please know that I have carefully considered each of your suggestions and incorporated them into the revised version of my paper.

If you have any additional comments or recommendations, I would be more than happy to address them. Your continued engagement and expertise are highly valued.

Thank you once again for your invaluable contribution to my work. I look forward to your future insights and the opportunity to collaborate again.

With heartfelt appreciation,

Yu Lyu